# Long-Smoldering T-prolymphocytic Leukemia: A Case Report and a Review of the Literature

Hilde K. Gjelberg [1], Lars Helgeland [1,2], Knut Liseth [3], Francesca Micci [4], Miriam Sandnes [5], Hege G. Russnes [6,7,8] and Håkon Reikvam [5,9,*]

[1] Department of Pathology, Haukeland University Hospital, N-5021 Bergen, Norway; hilde.gjelberg@helse-bergen.no (H.K.G.); lars.helgeland@uib.no (L.H.)

[2] Department of Clinical Science, University of Bergen, N-5021 Bergen, Norway

[3] Department of Immunology and Transfusion Medicine, Haukeland University Hospital, N-5021 Bergen, Norway; knut.liseth@helse-bergen.no

[4] Section for Cancer Cytogenetics, Institute of Cancer Genetics and Informatics, Oslo University Hospital, N-0424 Oslo, Norway; francesca.micci@ous-research.no

[5] Department of Medicine, Haukeland University Hospital, N-5021 Bergen, Norway; miriam.sandnes@uib.no

[6] Department of Pathology, Oslo University Hospital, N-0424 Oslo, Norway; h.e.g.russnes@medisin.uio.no

[7] Department of Cancer Genetics, Institute for Cancer Research, Oslo University Hospital, N-0424 Oslo, Norway

[8] Institute for Clinical Medicine, Faculty of Medicine, University of Oslo, N-0424 Oslo, Norway

[9] Department of Medical Science, University of Bergen, N-5021 Bergen, Norway

*   Correspondence: hakon.reikvam@uib.no; Tel.: +47-55975000

**Abstract:** T-prolymphocytic leukemia (T-PLL) is a rare malignancy of mature T-cells with distinct clinical, cytomorphological, and molecular genetic features. The disease typically presents at an advanced stage, with marked leukocytosis, B symptoms, hepatosplenomegaly, and bone marrow failure. It usually follows an aggressive course from presentation, and the prognosis is often considered dismal; the median overall survival is less than one year with conventional chemotherapy. This case report describes a patient with T-PLL who, after an unusually protracted inactive phase, ultimately progressed to a highly invasive, organ-involving disease. After initial treatments failed, a novel treatment approach resulted in a significant response.

**Keywords:** T-prolymphocytic leukemia (T-PLL); TCL1; ATM; JAK/STAT; alemtuzumab; pentostatin

## 1. Introduction

T-prolymphocytic leukemia (T-PLL) is a very rare neoplasm of T-cells with a mature, post-thymic immunophenotype and distinct clinical, cytomorphological, and cytogenetic features [1]. The disease usually has an aggressive course, with a median survival of less than one year with conventional chemoimmunotherapy [2]. T-PLL primarily occurs in the elderly, with the median age at presentation ranging from 62 to 69 years in different series [2–6]. It also typically presents with marked leukocytosis (>100 × 10⁹/L [reference range, 3.5–11.0]), anemia, thrombocytopenia, constitutional B symptoms, hepatosplenomegaly, and generalized lymphadenopathy. Cutaneous involvement and serous effusions occur at rates of approximately 20% and 15%, respectively [2]. Peripheral edema, particularly periorbital and/or conjunctival, occurs relatively frequently [7], while central nervous system (CNS) involvement is rare. T-PLL is a sporadic disease of unknown pathogenesis, although a possible link between EBV infection and T-PLL has been described [8].

T-PLL may initially present with an inactive, stable phase, although nearly all cases progress to active disease, usually within 1–2 years. The median survival after progression is as short as in the initially aggressive disease, and response to therapy is often poor. We present a case of T-PLL with an unusually long inactive phase that exceeded seven years before developing a more aggressive clinical course, and which ultimately responded to treatment.

## 2. Case

A 68-year-old male without previous significant medical history was seen in the outpatient clinic, to which he was referred due to incidentally detected leukocytosis. Apart from intermittent night sweats and slight fatigue, he was in good general health. The physical examination was unremarkable; notably, there were no general lymphadenopathy, skin lesions, or palpably enlarged spleen or liver. Laboratory tests revealed a white blood cell (WBC) count of $16.1 \times 10^9$/L, with an absolute lymphocyte count of $10.0 \times 10^9$/L (reference range, 0.7–5.3). The hemogram was otherwise normal, with a hemoglobin level of 15.7 g/dL (reference range, 13.4–17.0) and a platelet count of $173 \times 10^9$/L (reference range, 145–348). Blood levels of electrolytes and biochemical parameters such as uric acid, bilirubin, and creatinine were all within normal range. Serum lactate dehydrogenase (LDH) was normal at 155 U/L (reference range, 105–205). A computed tomography (CT) scan of the chest, abdomen, and pelvis was unremarkable; no lymphadenopathy, hepatosplenomegaly, or tumors were detected.

Examination of the peripheral blood smear showed monotonous, small- to medium-sized lymphoid cells—somewhat larger than typical chronic lymphatic leukemia (CLL) cells—with round nuclei (Figure 1a), some of which demonstrated a nucleolus. Peripheral blood flow cytometry analysis indicated an expanded population of lymphocytes, which accounted for 52% of viable WBC, showed a mature T-cell immunophenotype (positive for CD45 and pan-T antigens CD2, CD3, CD5, and CD7), and of which 90% were CD4-positive. The T-cell population further expressed CD26, CD28, and CD52. Among the negative markers were CD8, CD10, CD25, CD30, CD56, CD57, cyPerforin, cyGranzyme, and cyTCL1.

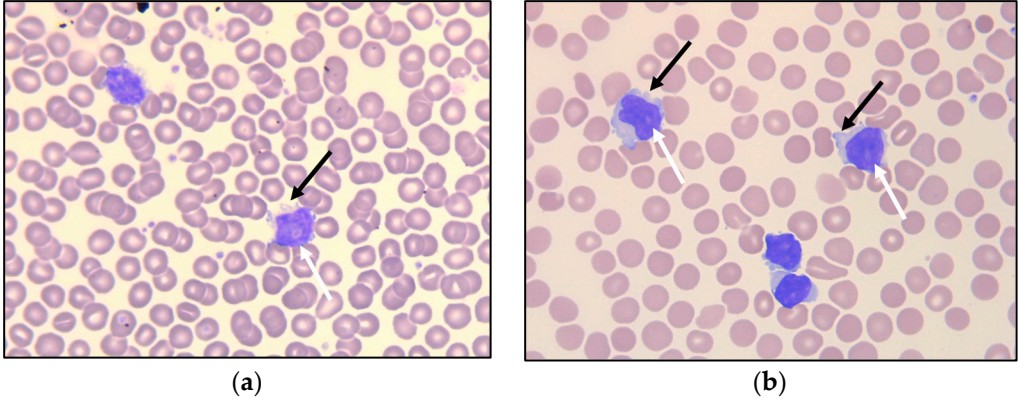

|                      (a)                      |                      (b)                      |

**Figure 1.** Peripheral blood smear at referral (**a**) and after 6.5 years of observation (**b**), with small- to medium-sized lymphoid cells having round nuclei, some with nucleoli (white arrows). Some of the lymphoid cells demonstrate small cytoplasmatic protrusions ("blebs"; black arrows).

The patient underwent a bone marrow aspiration and biopsy. The aspirate smears demonstrated lymphoid cells equivalent to the circulating cells in the peripheral blood. The biopsy demonstrated a slightly hypocellular bone marrow, with subtle interstitial infiltrates of small, CD3-positive lymphoid cells without significant atypia (Figure 2). The T-cells accounted for about 15–20% of the total nucleated bone marrow cells, with only small, loose aggregates. The majority of the T-cells were CD4-positive, but negative for CD57, Granzyme B, terminal deoxynucleotidyl transferase (TdT) and CD30, and for B-cell markers CD20, PAX5, and CD79a. There was no significant increase in reticulin fibers.

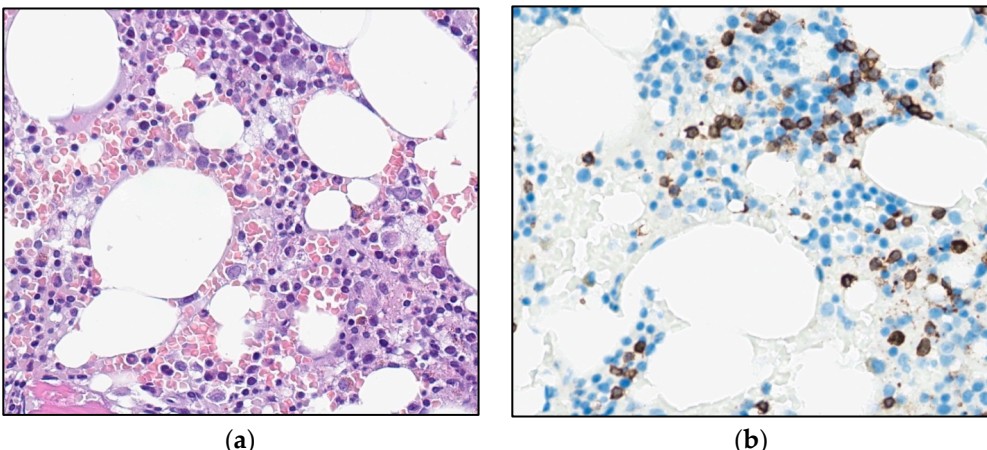

**Figure 2.** Photomicrograph of the bone marrow at referral: (**a**) H&E stain and (**b**) immunohistochemical staining for T-cell marker CD3 (32×).

These findings were considered insufficient for an unequivocal diagnosis of malignancy. However, polymerase chain reaction (PCR)-based clonality tests performed on the bone marrow aspirate demonstrated monoclonal rearrangement of the T-cell receptor (TCR)-γ chain gene. A diagnosis of clonal T-cell lymphocytosis was made, because the immunophenotype and cytomorphology of the T-cells excluded T-cell and NK-cell large granular lymphocytic leukemia (LGLL), and the clinical presentation and cytomorphology were not consistent with Sézary syndrome. The patient had no cytopenias or significant clinical symptoms; active treatment was therefore put on hold, and a watch-and-wait strategy with repeat checkups every three to six months was initiated. Apart from a brief hospitalization when the patient was diagnosed with a pulmonary embolism and successfully treated with initial and long-term anticoagulation, he remained well for more than six years, exhibiting a very slow increase in WBC counts (Figure 3).

Six and a half years after discovering the initial T-cell lymphocytosis, a more rapid increase in the WBC count was observed (Figure 3). At the same time, mild thrombocytopenia (platelet count $127 \times 10^9$/L) developed, along with a slight decrease in hemoglobin (14.1 g/dL) and an increase in LDH (325 U/L). The patient remained largely asymptomatic. However, six months later, the WBC count was $140 \times 10^9$/L and rapidly increasing (Figure 3), and the patient reported drenching night sweats, slight fever, and worsening fatigue.

Peripheral blood flow cytometry (Figure 4) revealed that the lymphoid population now constituted 92% of viable leukocytes and still demonstrated a mature, CD4-positive T-cell immunophenotype (positive antigens: CD3, CD4, CD5, CD7, CD26, CD28, CD45, CD45ro; among the negative antigens were CD10, CD56, CD99, and cyTCL1). A small subset was CD8-positive, and notably, CD52 was strongly positive. With the exception of the negative T-cell leukemia/lymphoma 1 (TCL1) stain, the immunophenotype was consistent with T-PLL.

As there were signs of incipient bone marrow failure, a new bone marrow biopsy was performed, demonstrating increased cellularity compared with the biopsy seven years earlier. Interstitial and diffuse infiltrates of small T-cells now comprised 30% of the nucleated bone marrow cells, accompanied by a diffuse increase in reticulin fibers (Figure 5a,b). An expanded immunohistochemical panel showed a corresponding immunophenotype to that seen in the flow cytometry. The assessment of TCL1 was technically difficult due to an increased number of B-cells admixed (Figure 5c–f), although it was interpreted as negative. Molecular clonality assessment demonstrated similar clonal peaks as those seen initially, and peripheral blood smears demonstrated atypical lymphoid cells, with a round nucleus and a prominent nucleolus (Figure 1b).

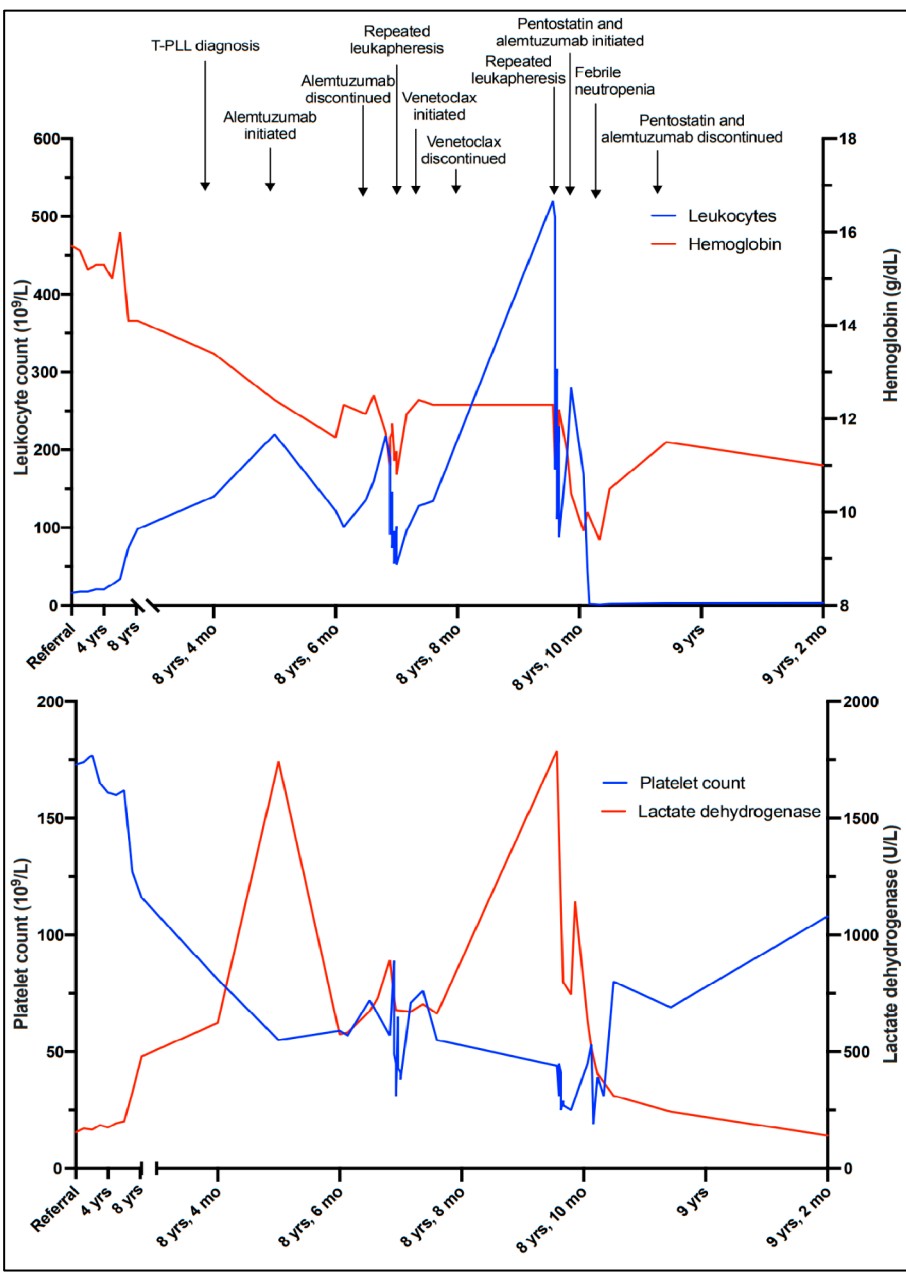

**Figure 3.** Timeline illustrating the leukocyte count, hemoglobin level, platelet count, and lactate dehydrogenase level corresponding to clinical events.

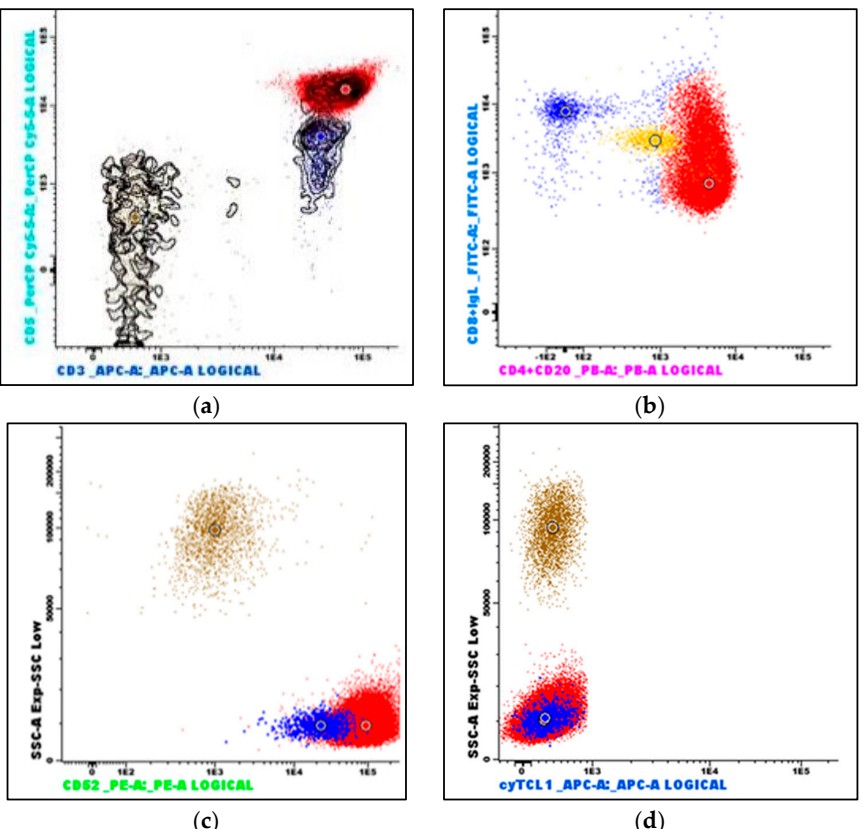

**Figure 4.** Flow cytometry immunophenotyping of peripheral blood: (**a**) The neoplastic T-cells demonstrate bright CD3 and CD5 expression (red dots); reactive T-cells demonstrate dimmer CD5 expression (blue dots). (**b**) The neoplastic T-cells are predominantly CD4-positive, a subset is CD8-positive, and they are strongly CD52-positive (**c**). (**d**) The neoplastic T-cells are TCL1-negative. (Brown dots are granulocytes, and yellow dots are monocytes. The larger dots indicate the median value of the population with the identical color).

Cytogenetic analysis confirmed a complex karyotype with the inversion of chromosome 14, inv(14) (q11q32), trisomy 8, and rearrangements of chromosome 11 del(11) (q23), among other abnormalities (Figure 6). Fluorescence in situ hybridization (FISH) showed a TCRA/D split (mapping to 14q11), but no involvement of the ATM locus (mapping to 11q22). With all the findings taken into consideration, a diagnosis of T-PLL was made, as all three major and most of the minor (three out of four) diagnostic criteria established by the T-PLL International Study Group (TPLL-ISG) were fulfilled (Table 1) [9].

**Table 1.** Requirements for establishing the diagnosis of T-PLL (TPLL-ISG, 2019 [9]).

| Major Criteria | Minor Criteria (at Least 1 Required) |
|---|---|
| >5 × $10^9$/L cells of T-PLL phenotype in peripheral blood or bone marrow | Abnormalities involving chromosome 11 (11q22.3; ATM) |
| T-cell clonality (by PCR for TRB/TRG, or by flow cytometry) | Abnormalities in chromosome 8: idic(8)(p11), t(8;8), trisomy 8q |
| Abnormalities of 14q32 or Xq28 OR expression of TCL1A/B, or MTCP1 * | Abnormalities in chromosome 5, 12, 13, 22, or complex karyotype |
| | Involvement of T-PLL specific site (e.g., splenomegaly, effusions) |

* Cases without TCL1A, TCL1B, MTCP1 rearrangement, or their respective overexpression, are collected as TCL1-family negative T-PLL.

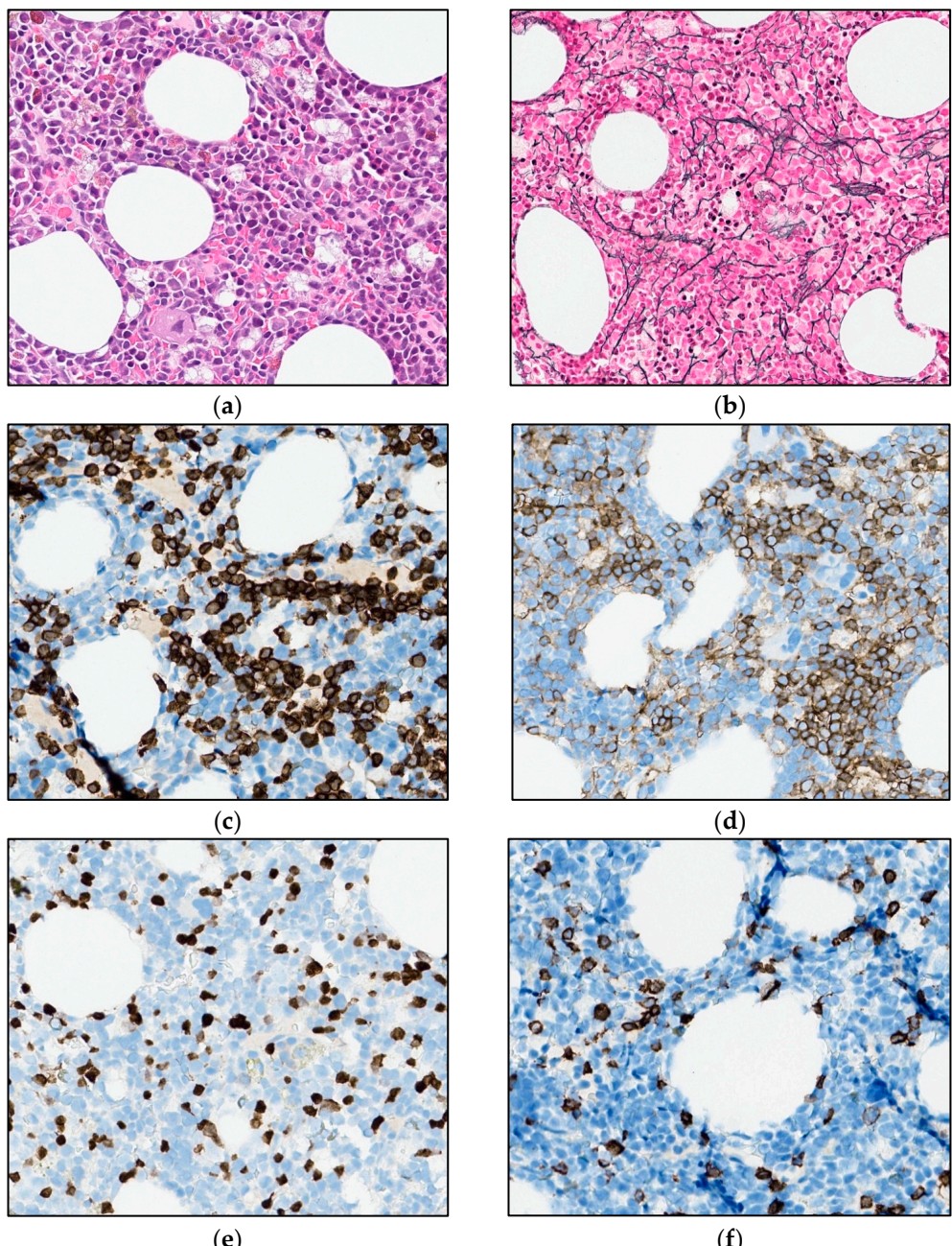

**Figure 5.** Photomicrograph of the bone marrow after 6.5 years of observation: (**a**) H&E (32×). (**b**) Reticulin stain (25×). (**c**,**d**) Immunohistochemical stains for CD3 and CD4 (32×). (**e**,**f**) Immuno-histochemical stains for TCL1 and CD20 (32×).

A CT scan demonstrated an enlarged spleen (12 × 17 × 19 cm), a slightly enlarged liver, enlarged mesenteric lymph nodes, and pleural effusion. Furthermore, he developed maculopapular erythematous lesions on his legs (Figure 7a); a skin biopsy demonstrated dermal perivascular and periadnexal infiltrates of atypical T-lymphocytes (Figure 7b) with sparing of the epidermis, consistent with cutaneous involvement of T-PLL.

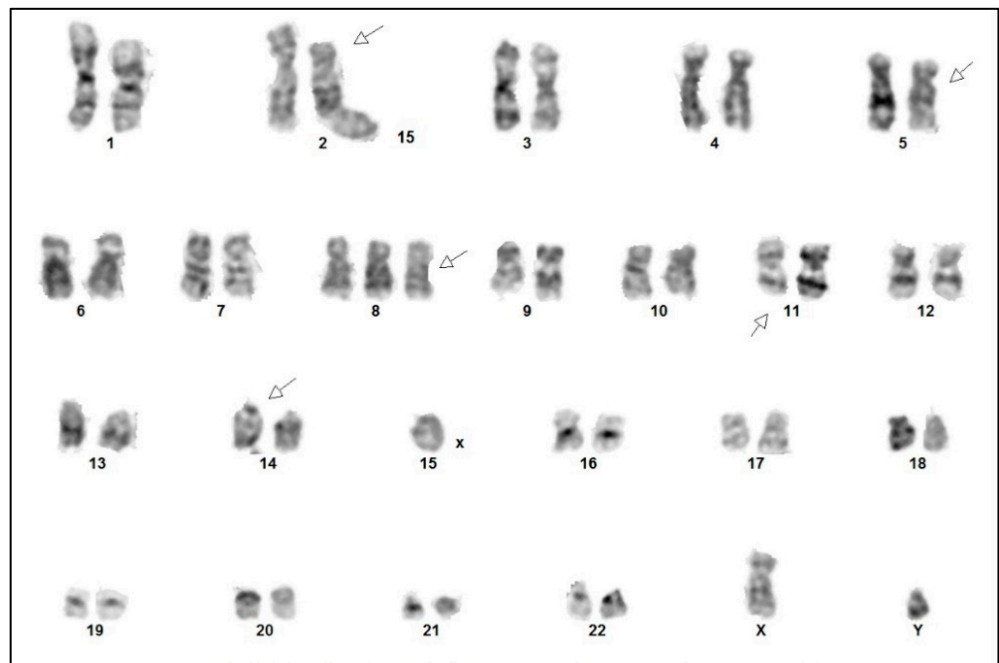

**Figure 6.** Cytogenetic analysis of the tumor cells demonstrating a complex karyotype: 47,XY,del(2)(p21),del(5)(q11q13),+8,del(11)(q23),inv(14)(q11q32).

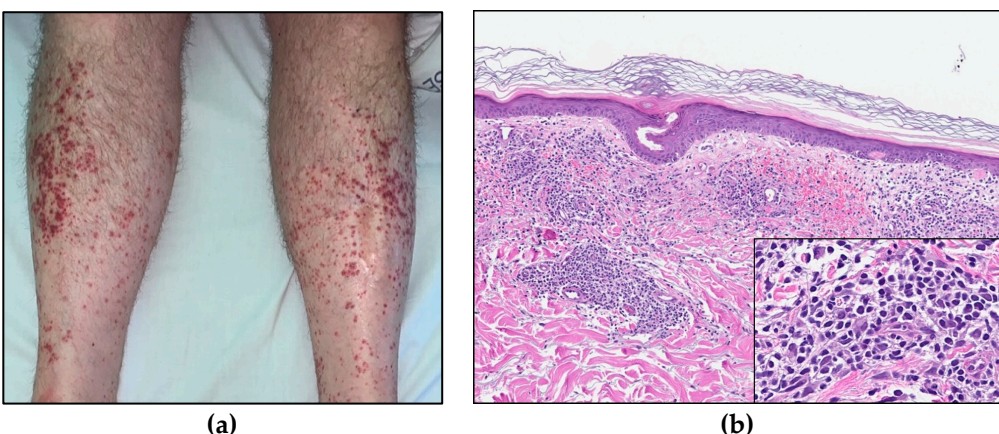

| **(a)** | **(b)** |

**Figure 7.** (**a**) Maculopapular skin lesions on both lower extremities. (**b**) Skin biopsy (HE, 8× and insert 40×).

Due to signs of bone marrow failure and increasing B symptoms there was indication for treatment, and monotherapy with alemtuzumab was initiated. In parallel with tumor lysis prophylaxis and induction therapy, alemtuzumab was administered with gradual dose escalation to a maintenance dosage of 30 mg three times weekly, and the treatment was well tolerated. Initially, a drop in the WBC count was observed, from $240 \times 10^9$/L to $110–120 \times 10^9$/L (Figure 3). However, values stabilized at $>100 \times 10^9$/L despite several cycles of treatment, and partial remission was not achieved. He remained thrombocytopenic, with platelet counts between 50 and $60 \times 10^9$/L. As further alemtuzumab treatment in monotherapy was considered ineffective, it was discontinued after 17 infusions. To investigate the possibility of experimental treatment, and possible inclusion in the IMPRESS-Norway study (a national prospective, non-randomized clinical trial evaluating the efficacy of anti-cancer drugs on new indications [10]), tumor material (peripheral blood) was submitted for molecular profiling with a 523-gene panel analysis (TruSight Oncology 500). The results revealed a low tumor mutational burden (7.9 mutations/Mb). A somatic point mutation in the JAK3 gene, (A573V), was also detected.

Without active treatment, the patient experienced rapidly increasing WBC counts and developed symptoms of hyperviscosity, including headaches, lethargy, and dyspnea. Several rounds of therapeutic leukapheresis were performed, resulting in an almost 50% reduction in WBC count (Figure 3). A second-line, off-label treatment with venetoclax (an oral inhibitor of the BCL-2 (B-cell lymphoma 2) protein encoded in humans by the BCL-2 gene) was approved and initiated, with gradual dose escalation up to 400 mg daily. Unfortunately, there was no response to this treatment, and the patient's WBC counts continued to rise, accompanied by the worsening of his symptoms and cytopenias. He developed renal failure and dyspnea due to pleural effusions, and he was hospitalized for several weeks while repeated leukapheresis procedures and drainage of the pleural fluid were performed. Based on a review of clinical trials, recommended treatment algorithms, and available pharmacological interventions [11,12], a third-line treatment with pentostatin in combination with alemtuzumab was initiated. Alemtuzumab was given in an escalating dosage, and pentostatin 4 mg/m$^2$ intravenous was given once a week for four weeks, and then every second week. After about two weeks, his WBC count started to decrease and his bone marrow function improved, showing an increase in platelet and hemoglobin levels. On day 17, he developed neutropenia ($1.2 \times 10^9$/L) and fever consistent with febrile neutropenia, along with respiratory symptoms. A chest radiograph established the presence of pneumonia. He was successfully treated with antibiotic therapies (piperacillin/tazobactam). Apart from this, he tolerated the treatment well, exhibiting a further decrease in his WBC count until it stabilized at around $3.0 \times 10^9$/L, and continued improvement of his general condition. The treatment was terminated after six weeks. The patient is currently, eight and a half years after his initial referral, doing well without active treatment and without sign of leukocytosis.

## 3. Discussion

This case report presents a patient with T-PLL who experienced a very long time of inactive T-cell lymphocytosis, which finally evolved into a more aggressive disease course but ultimately responded to a third line of treatment with pentostatin-based therapy in combination with alemtuzumab.

T-PLL is a rare malignancy of mature T-cells. The diagnosis is made based on a multidisciplinary approach, integrating clinical data with the morphological findings in peripheral blood smears and/or bone marrow aspirates, as well as the immunophenotype and the cytogenetic features of the abnormal T-cells. The T-PLL cell is typically a small-to medium-sized lymphoid cell exhibiting the features of prolymphocytes, i.e., cells with non-granular, basophilic cytoplasm, often having small cytoplasmatic protrusions, and an enlarged round to oval, or irregularly shaped nucleus with a single distinct nucleolus. Less common morphological variants include a small-cell variant (20%) and a variant with irregular nuclei similar to Sézary cells (cerebriform variant; 5%) [9]. Bone marrow biopsy typically demonstrates interstitial and/or diffuse infiltration, often associated with an increase in reticulin fibers. Affected lymph nodes demonstrate an expanded paracortical zone and infiltration of the atypical T-cells. In the spleen, there is significant involvement of both red and white pulpa. Cutaneous lesions demonstrate perivascular and periadnexal infiltrates of lymphoid cells, typically without epidermotropism.

The neoplastic T-cells demonstrate a mature, post-thymic immunophenotype via flow cytometry and/or immunohistochemistry, i.e., negativity for immature T-cell markers such as TdT, CD34, and CD1a, and positivity for pan-T antigens CD2, CD3, CD5, and CD7. CD7 is expressed with high intensity. The vast majority are CD4-positive [2], of which a subset coexpresses CD4 and CD8. A CD4-/CD8+ phenotype or a double-negative (CD4−/CD8−) phenotype is less frequent. CD52 is typically strongly expressed, which is of therapeutic significance, as it may be targeted by the monoclonal CD52 antibody alemtuzumab [13,14].

The T-cell receptor (TCR) β and/or γ chain genes are rearranged in T-PLL and can thus be used to demonstrate the clonal nature of the T-prolymphocytes. T-PLL cells typically demonstrate complex chromosomal abnormalities with recurrent genetic features [9]. More

than 90% harbor rearrangements involving the TCL1 (T-cell leukemia/lymphoma 1) A/B locus and the TCR $\alpha/\delta$ locus, both on chromosome 14 and seen either as inv(14)(q11q32) or t(14;14)(q11;q32). These result in the activation of the oncogene TCL1 and subsequent TCL1 protein overexpression, a diagnostic hallmark for T-PLL. Rare cases of T-PLL harbor rearrangements involving the TCL1 homolog MTCP1 (Mature T-cell Proliferation 1) on the X chromosome through a t(X;14)(q28;q11). A small subset of cases that otherwise meet the diagnostic criteria for T-PLL do not carry TCL1 rearrangements and consequently produce no TCL1 protein overexpression, resulting in TCL1-family negative T-PLL.

Activation of TCL1 or MTCP1 is assumed to be the primary genetic event in T-PLL; however, this is not sufficient to drive leukemogenesis alone [15]. Other frequently occurring genetic alterations include inactivating mutations or deletions in the DNA repair master regulator ATM (ataxia telangiectasia mutated), which are observed in >80% of T-PLL cases [16]; downstream dysregulation of p53; and abnormalities in chromosome 8, including trisomy 8q [9]. Comprehensive genome-wide analysis of T-PLL cases has further demonstrated alterations in epigenetic regulators like EZH2, KMTs and HDACs and DNA repair/checkpoint proteins (CHEK2), as well as a prominent role for activating mutations affecting the interleukin-2 receptor gamma (IL2RG)-JAK-STAT axis [17]. Activating mutations of JAK3 have been described in up to one-third of T-PLL cases in different series [15,18], and have shown a significant negative impact on overall survival [18].

T-PLL must be separated from other mature T-cell malignancies with leukemic presentation, such as adult T-cell leukemia/lymphoma (ATLL), T-cell large granular lymphocytic leukemia (T-LGLL), NK-cell large granular lymphocytic leukemia (NK-LGLL), and Sézary syndrome. Serology for the human T-lymphotropic virus-1 (HTVL1) should be negative in order to make ATLL unlikely. The cytomorphology, immunophenotype, and clinical presentation helps exclude T-LGLL (granular lymphocytes in smears, typically CD8+ and CD57+), NK-LGLL (usually surface CD3− and CD56+), and Sézary syndrome (cerebriform nuclei, characteristic presentation with erythroderma, often loss of one or more pan-T markers). These entities also lack the recurrent genetic features typical of T-PLL. In 2019, the TPLL-ISG presented standardized criteria for diagnosis, treatment indication, and response evaluation (Table 1) [9].

Initially, up to 30% of T-PLL cases present as inactive stable disease, but within 1–2 years nearly all cases progress to active disease [9]. Markers of active disease and indications for treatment are disease-related B symptoms, rapidly increasing lymphocytosis, bone marrow failure, skin infiltration, and rapidly enlarging lymph nodes, spleen, and/or liver [19]. In the inactive phase, observation of blood counts and clinical examination at regular intervals is recommended, although early treatment has not been shown to improve clinical outcomes. Hence, in the initial phase, our patient was followed with a watch-and-wait approach.

In patients with active T-PLL, the disease progresses quickly without treatment, and conventional chemotherapy has proven ineffective. The introduction of the anti-CD52 antibody alemtuzumab in the last two decades has significantly improved outcomes, with reported response rates ranging from 80 to 100% and 50 to 76% in patients with previously untreated and treated T-PLL, respectively. Complete remission is seen in 40 to 100% of patients [2]. Unfortunately, the treatment response is not lasting, and the disease will inevitably progress. The median duration of remission for responders is less than 2 years [9]. For patients with suboptimal responses to alemtuzumab, the addition of the purine analogue pentostatin may be beneficial [11,19]. In the setting of relapsed or refractory disease, bendamustine or cladribine show moderate clinical effects [20]. The BCL-2 inhibitor venetoclax (ABT 199) has demonstrated clinical response in late-stage refractory T-PLL [21]. However, the median overall survival with modern therapy is 21 months [6]. Allogeneic hematopoietic stem cell transplantation (allo-HSCT) remains the only potentially curative treatment for T-PLL, and patients who achieve complete remission should be considered for allo-HSCT [9]. Due to the advanced age of most of the patients, only about 40% are eligible for allo-HSCT, and transplant-related mortality

and relapse rates remain high [7,22]. Promising new approaches include novel agents such as the BCL-2 family inhibitor ABT-737 and JAK/STAT inhibitors [9]. Implementations of immune-checkpoint inhibitors or CAR-T cell therapy are at the stage of pre-clinical assessments of activity and feasibility [20]. Our patient experienced limited effects from alemtuzumab and venetoclax in monotherapy, and only when pentostatin was added did he respond to the treatment. This clearly illustrates the potential benefits of pentostatin in refractory cases of T-PLL, potentially mediated by synergistic effects between pentostatin and alemtuzumab [11,12].

Our case represents an unusual course of a very rare disease: a TCL1-negative T-PLL with an unusually long inactive phase. Patients with T-PLL with a prolonged inactive phase have been reported [5,23]. However, for the vast majority of these patients, the duration of the inactive phase was less than 2 years [9]. The "indolent" clinical presentation and lack of TCL1 expression made it challenging to arrive at the correct diagnosis. Thus, the question must arise: does our case represent a "true" T-PLL? Barring the lack of TCL1 expression, the cytomorphological, immunophenotypical, and clinical findings were found to be consistent with T-PLL, and the characteristic cytogenetic aberrations associated with T-PLL were demonstrated (inversion on chromosome 14 (inv(14)(q11q32), trisomy 8, and abnormalities of chromosome 5). Thus, all three major and three out of four minor TPLL-ISG diagnostic criteria were fulfilled (Table 1).

Cytogenetic findings showed rearrangement of chromosomal band 14q32 (the band where the TCL1 gene is located) through an inversion 14. However, the FISH experiment using a locus-specific probe for TCL1 showed no rearrangement of the gene. This is in contrast with the findings by Sun et al., who identified TCL1 rearrangements in all patients with 14q32 abnormalities [24]. Furthermore, our case showed lack of TCL1 expression by flow cytometry and immunohistochemistry, something that may be due to alterations at the protein level such as the presence of a truncated protein. Previous studies have shown that high TCL1 expression, measured using flow cytometry and/or immunohistochemistry, is associated with poorer outcomes [4]. Herling et al. found that T-PLL cases that were negative or low for TCL1, including some carrying a chromosome 14q32 rearrangement, showed lower WBC counts. Thus, the lack of TCL1 overexpression may in part explain the initial, indolent clinical course in our case.

There have been some reports of T-PLL with a longer stable phase, in most cases of less than 5 years [3,5]. The duration of the inactive phase is not affected by treatment [5]. Most of the patients will have none or few symptoms in this phase, raising the question: is there a silent preleukaemic state preceding most T-PLL cases?

Given the low incidence of T-PLL, there is still a major gap in knowledge regarding disease biology, the consensus on correct diagnostic criteria, and optimal treatment choices. There appears to be a heterogeneity within disease biology for T-PLL, which is reflected in treatment response and prognosis. This is still not fully understood or explored. We therefore propose future multicenter studies mapping these factors, which can create a foundation for future treatment algorithms to improve prognosis for this patient group.

**Author Contributions:** Conceptualization, H.K.G. and H.R.; methodology, H.K.G., L.H., K.L., F.M., M.S., H.G.R., and H.R.; software, H.K.G., M.S., H.G.R., and H.R.; validation, H.K.G., L.H., and H.R.; formal analysis, H.K.G. and L.H.; investigation, H.K.G., L.H., K.L., F.M., M.S., H.G.R., and H.R.; resources, H.K.G., H.G.R., and H.R.; data curation, H.K.G., L.H., K.L., F.M., and M.S. All authors have read and agreed to the published version of the manuscript.

**Funding:** This research received no external funding.

**Institutional Review Board Statement:** Not available.

**Informed Consent Statement:** Informed consent was obtained from the patient for publication of this case report.

**Data Availability Statement:** Data is maintained within this paper.

**Conflicts of Interest:** The authors declare no conflict of interest.

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
