# Peer review of "Long-Smoldering T-prolymphocytic Leukemia: A Case Report and a Review of the Literature"

_curroncol, doi:10.3390/curroncol30110727_

Round 1

Reviewer 1 Report

Comments and Suggestions for Authors

This is a very interesting case of TCL-1 negative T-PLL with a long-standing prodromal phase. This is very relevant as clonal T-cell populations with minor cytological atypia are detected occasionally in the peripheral blood. The significance of those is not understood, and this report highlights the importance of follow-up clinical/laboratory evaluation in these patients.

The report is well-written, the quality of the images is good, and the discussion section is well-crafted. 

Only minor changes are requested:

1) Can you expand on the flow cytometry immunophenotype detected at the initial presentation? Any immunophenotypical aberrancies were detected? Similarly, can you expand on the immunophenotype after six years (only CD4 and CD52 are included)?

2) Can you include pictures of the T-PLL lymphocytes in the peripheral blood, when overt disease was present?

Reviewer 2 Report

Comments and Suggestions for Authors

The case report " Long smoldering T-prolymphocytic leukemia: A case report and a case-based review of the literature" provides valuable insights into the diagnosis and treatment of a rare and challenging malignancy. They highlighted a case with an unusually long inactive phase of T-prolymphocytic leukemia (T-PLL) and the patient's response to novel treatment strategies. However, there are some issues and questions that should be addressed in the manuscript.

1-  Despite such evaluation by the authors, "To the best of our knowledge, no other T-PLL case with a similarly long inactive phase has been reported", 25 patients with long initial indolent clinical course with stable moderate leucocytosis were reported in only 2002 (https://doi.org/10.1046/j.1365-2141.1998.00977.x and 10.1002/ccr3.1528 )

2-  Previous published case reports on clinicopathologic features of T-prolymphocytic leukemia can be tabulated.

3- While the report mentions the importance of distinguishing T-PLL from other mature T-cell malignancies, it could benefit from a more detailed discussion of the key differential diagnoses, including Adult T-cell Leukemia/Lymphoma (ATLL), T-cell Large Granular Lymphocytic Leukemia (T-LGLL), NK-cell Large Granular Lymphocytic Leukemia (NK-LGLL), and Sézary Syndrome. Providing a brief comparison of these conditions and explaining how the presented case was differentiated from them would be helpful for readers.

4- The case report mentions that the patient responded well to third-line treatment with pentostatin in combination with alemtuzumab, but it could delve deeper into the rationale behind choosing this treatment regimen. Discussing the mechanisms of action of these drugs and why they were chosen for this specific patient could add depth to the report.

5- It could be beneficial to conclude the report with a brief discussion of potential future directions in T-PLL research and treatment. This might include emerging therapies, ongoing clinical trials, or areas where further investigation is needed.

6- Correct "109/L" to "109/L " throughout manuscript

Comments on the Quality of English Language

Moderate editing of English language required

Round 2

Reviewer 1 Report

Comments and Suggestions for Authors

The authors have addressed all the comments. 

Author Response

Thank you again for taking the time to review this manuscript.

Reviewer 2 Report

Comments and Suggestions for Authors

I am satisfied that the authors have addressed mostof my previous concerns about the article. However, I think it would be more appropriate for the authors to include a table summarizing previously published case reports on the clinicopathological features of T-prolymphocytic leukemia.

Comments on the Quality of English Language

Minor editing of English language required

Author Response

Thank you again for your valuable feedback. We fully agree that generating a table summarizing previously published case reports on the clinicopathological features of T-prolymphocytic leukemia would be interesting and worthwhile. However, this would require a systematic literature review of published case studies. It seems that this work is so extensive that it could form the basis for a separate and independent work. Unfortunately, this is currently beyond our capacity, especially with regard to the short time frame given.

A colleague fluent in English writing has proofread our manuscript, and corrections are highlighted.